# Unlocking [CLS] Features for Continual Post-Training

**Murat Onur Yildirim, Elif Ceren Gok Yildirim, Joaquin Vanschoren**
*AMOR/e Lab, Eindhoven University of Technology*
*Correspondence to: m.o.yildirim@tue.nl*

**Reviewed on OpenReview:** *https://openreview.net/forum?id=OWfWyj6krc*

## Abstract

Continual learning requires models to integrate new classes or domains over time while preserving previously acquired knowledge. Within this paradigm, foundation models often achieve strong performance, but they still remain subject to the stability–plasticity trade-off, where excessive plasticity leads to forgetting of prior knowledge, and excessive stability constrains the adaptation. This necessitates an effective post-training strategy that introduces minimal yet functional modifications. To address this challenge, we first introduce a new parameter-efficient fine-tuning module 'Learn and Calibrate', or LuCA, designed to acquire task-specific knowledge through an adapter-calibrator couple, enabling well-refined feature representations. Then, for each task, we deploy a sparse LuCA module on top of the last classification token [CLS] just before the classifier, which we refer to as 'Token-level Sparse Calibration and Adaptation', or TOSCA. By leaving the generalization capabilities of the foundation models intact and adapting exclusively via the last token, our approach achieves a harmonious balance between stability and plasticity while reducing both training and inference complexity. We demonstrate that TOSCA yields state-of-the-art performance while introducing ~8× fewer parameters compared to prior methods. The code is available at https://github.com/muratonuryildirim/tosca.

## 1 Introduction

Learning continuously from a series of concepts or classes using a unified model is a challenging problem due to catastrophic forgetting (1), a phenomenon where the model's performance on earlier concepts degrades as new classes or domains are observed. Class-incremental learning (CIL), a branch of continual learning, addresses this issue by enabling models to acquire knowledge from new classes while preserving their ability to correctly classify previously learned categories (2). Until recently, most CIL methods have focused on relatively small networks such as ResNets (3) and often trained them starting from random initialization (4; 5). With the rise of large Foundation Models (FMs) (6; 7; 8) such as Vision Transformers (ViTs) (9), many CIL methods now capitalize on the robust representations provided by FMs, marking a significant paradigm shift in the field, showing that leveraging strong initial representations from large-scale pre-training significantly enhances incremental learning (10; 11; 12). However, sequential fine-tuning of FMs inevitably alters the pre-trained representations, leading to substantial forgetting (13; 14; 15).

To tackle this, post-training adaptation strategies such as learnable prompts (16; 17; 18; 19; 20; 21) and lightweight adapters (22; 23; 24) restrict updates to small subsets of parameters, atop large frozen pre-trained backbones. While this helps to mitigate forgetting, they introduce new trade-offs. Learnable prompts aim to steer the FMs existing knowledge to new tasks by introducing small trainable embeddings, keeping the model's core parameters unchanged. This guides the model to activate relevant pre-existing knowledge for new tasks and offers great stability, but often limits task-specific adaptability. In contrast, adapters inserts small trainable neural networks directly into the FM's layers to provide localized feature refinement with high plasticity, but this flexibility often comes at the cost of quadratic parameter growth with increasing model depth. This trade-off exemplifies the well-known stability–plasticity dilemma (25) and motivates the central question of this work:

> *'How can we efficiently tackle the stability-plasticity dilemma in continual post-training?'*

To address this question, we take inspiration from neuroscience, where the brain achieves continual learning by forming invariant representations in the ventral visual stream (26; 27), while flexibly adapting and modulating them through task-specific circuits in the prefrontal cortex (28; 29; 30). In other words, the prefrontal cortex receives these stable visual representations and refines them through selective synaptic plasticity, enabling flexible adaptation to task demands and effectively guiding appropriate behavioral responses.

Analogously, we aim to leverage a large pre-trained model to emulate the ventral visual stream, which extracts stable and invariant features. To adapt these general features for specific tasks, we insert small lightweight modules just before the decision layer, mirroring how cortical circuits flexibly refine representations based on task-specific demands. This design avoids redundant relearning of low-level features and aligns with biological learning principles, enhancing both efficiency and adaptability.

To this end, we first introduce a new parameter-efficient fine-tuning (PEFT) module 'Learn and Calibrate', or *LuCA*, which comprises two components: (1) a residual adapter that applies task-specific feature transformations, and (2) a calibrator that reweighs and enhances the adapted features via attention-like gating.

Then, to enable post-training in CIL setup, we train a single sparse LuCA module for each task, operating exclusively on the final `[CLS]` token representation of ViTs. We refer to this approach as 'Token-level Sparse Calibration and Adaptation,' or briefly *TOSCA*. By localizing adaptations at the final semantic aggregation point and preserving the low/mid-level feature hierarchy, TOSCA mirrors the harmony between the ventral visual stream and the prefrontal cortex.

Specifically, task-specific information is residually acquired just before classification through a dedicated LuCA module, sparsified via $\ell_1$-regularization to promote parameter orthogonality, which improves the specialization and distinctiveness across modules. This targeted injection in a continual post-training protocol preserves the stability of the rich, generalizable features of the FMs, while providing the necessary plasticity through precise, task-specific adjustments at the point of decision-making. The inference protocol leverages entropy minimization over task-specific predictions, as correct modules produce low-entropy class distributions. This approach removes the reliance on task identifiers or exemplar replay while achieving state-of-the-art performance without complicated procedures.

Our contributions are three-fold:

I.   We introduce a new PEFT module LuCA designed to learn task-specific residual transformations while refining features through additional calibration gating.

II.  We propose TOSCA, a neuro-inspired and theoretically grounded continual post-training approach that strategically integrates our LuCA module at the final semantic aggregation point in the network. This balances stability and plasticity while maintaining a model-agnostic parameter count, unlike many prompt- and adapter-based methods that scale linearly with the number of layers.

III. We validate TOSCA's advantages with extensive experiments on six benchmarks. We find that TOSCA yields (i) 7–21% higher accuracy than prompt-based methods and 4–12% higher than adapter-based methods on out-of-distribution datasets, (ii) ~2.5× faster overall runtime, and (iii) ~8× fewer parameters than layer-wise adapters.

## 2   Related Work

**CIL with Randomly Initialized Models.**   Not a long time ago, the focus in CIL was training deep neural networks sequentially from scratch, and the strategies can be categorized into four main approaches. Regularization-based methods (31; 32; 33; 34) maintain the model by selectively stabilizing changes in parameters or predictions. Replay-based methods approximate and reconstruct previously learned data distributions by either storing (35; 36; 37; 38; 39; 40; 41; 42; 43) or generating (44; 45; 46; 47; 48) samples from past experiences. Architecture-based methods allocate distinct parameters and subspaces to different sets of classes, whether expanding the architecture (49; 50; 51; 52; 53) or pruning the existing one (54; 55; 56; 57; 58; 59; 60; 61) to obtain and preserve key parameters.

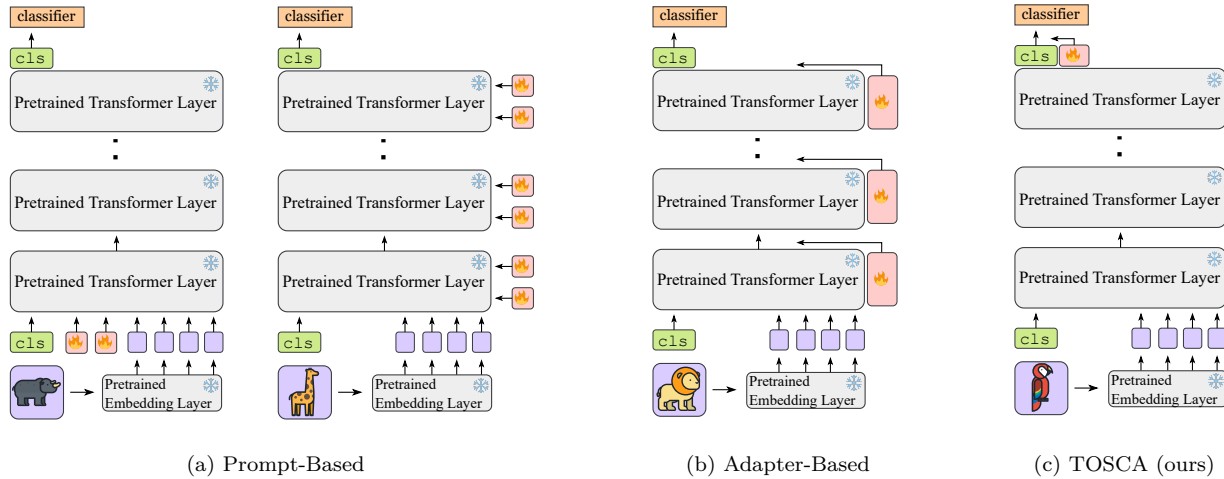

Figure 1: Overview of the FM-based continual post-training methods. Prompt-based methods influence the self-attention process of an FM, either from the input layer or across all layers. Adapter-based methods enable task-specific adaptations by inserting lightweight neural modules into the FM's layers. In contrast, we propose to train a single module that operates exclusively on the final `[CLS]` token representation, efficiently adapting and calibrating features just before classification. This design offers a streamlined and effective alternative to existing methods.

**CIL with Pre-Trained Models.** In contrast, recent advancements in CIL research have shifted towards leveraging pre-trained FMs since representations derived from those models have proven to be effective not only in facilitating knowledge transfer but also in mitigating catastrophic forgetting during downstream continual learning (10; 11) with minimal adaptations (12). Therefore, post-training methods in this context aim to improve performance with minimal additions and modifications while freezing the FMs. L2P (18) borrows a technique from NLP by introducing a learnable prompt pool and selecting instance-specific prompts via a key-query matching selection mechanism to guide the FMs response. DualPrompt (19) extends L2P by designing G-Prompt and E-Prompt, which encode task-invariant and task-specific instructions respectively. CODAPrompt (20) uses contrastive learning to decorrelate representations of the prompts to reduce interference and combine them by attention-based weighting method. HiDePrompt (21) leverages prompts together with class-level feature statistics to enable implicit feature-space replay and improve class-wise alignment. APER (22) explores various PEFT methods and shows that a simple prototypical classifier called Simple-CIL serves as a strong baseline. RanPAC (15) extends SimpleCIL by randomly projecting features into a high-dimensional space with an analytical ridge classifier. EASE (23) attaches adapters to each layer of FMs to create expandable subspaces, and during inference, it concatenates all feature representations from different sets of adapters. MOS (24) adds replay generation for classifier alignment and an adapter merging over EASE to reduce mistakenly retrieving irrelevant modules during inference due to parameter drift.

## 3 Background

In this section, we first formally introduce the preliminaries of the CIL and describe how FMs are utilized to facilitate incremental learning. We then provide an overview of existing approaches, highlighting their strengths and the key limitations they face in effectively addressing the challenges of continual learning.

### 3.1 Class-Incremental Learning (CIL)

Class-incremental learning is a scenario where a model continually learns to classify new classes to build a unified classifier (35). Formally, we train models sequentially on a series of tasks $\{\mathcal{T}_t\}_{t=1}^T$ where each task with a task-specific dataset $\mathcal{D}_t = \{(x_i, y_i)\}_{i=1}^{n_t}$ introduces a set of previously unseen classes $y_i \in Y_t$ with $n_t$ instances, ensuring $Y_t \cap Y_{t'} = \emptyset$ for any $t \neq t'$, i.e. non-overlapping classes for different datasets, while the underlying data distribution may evolve over time. During the $t$-th training stage, the model is updated using data exclusively from $D_t$.

From the model perspective, following typical FM-based CIL works (18; 19; 20; 22; 23; 24), we assume that a foundation model is available for the initialization of the model $f(\mathbf{x})$ which we define with two components: $f(\mathbf{x}) = W^\top \Phi(\mathbf{x})$, where $\Phi : \mathcal{X} \to \mathbb{R}^d$ is the feature extractor and $W \in \mathbb{R}^{d \times |\mathcal{Y}|}$ is the classifier. For a vanilla ViT (9), the initial embedding layer converts the input image into a sequence of token embeddings, denoted as $\mathbf{x}_e = [\mathbf{x}_{e_0}, \mathbf{x}_{e_1}, \ldots, \mathbf{x}_{e_L}] \in \mathbb{R}^{(L+1) \times d}$, where $L$ is the number of patch tokens and $\mathbf{x}_{e_0}$ represents the prepended [CLS] token. The sequence $\mathbf{x}_e$ is then processed through subsequent transformer layers, including multi-head self-attention and feed-forward networks, to produce the final embeddings. Following the standard practice, we use [CLS] token $\mathbf{x}_{e_0}$ as the global image representation and consider as $\Phi(\mathbf{x})$.

The effectiveness of the model is evaluated across all encountered classes, collectively can be represented as $\mathcal{Y} = Y_1 \cup Y_2 \cup \cdots Y_t$ after each learning stage. Specifically, we aim to find a model $f(\mathbf{x}) : \mathcal{X} \to \mathcal{Y}$ that minimizes empirical risk across all test dataset **without task indices** by balancing between learning new classes and retaining information about old ones in the **replay-free setting** (18; 19; 20; 22; 23).

### 3.2 Overview of Post-Training in CIL

In the era of FMs, the main idea of many works seeks to modify the pre-trained weights slightly with post-training, to maintain the generalization strength and we can mainly divide these approaches into three.

**Learning Prototypical Classifiers.** These methods (15; 17; 22) focus on learning a set of prototypical class representations, typically by computing class centroids or prototypes from the features of incremental classes. Given an input instance $\mathbf{x}$ with label $y \in Y_t$, let $\Phi(\mathbf{x})$ be its feature vector extracted by a pre-trained backbone. Then, the class prototype $\mathbf{p}_y$ is defined as

$$\mathbf{p}_y = \frac{1}{n_t} \sum_{i=1}^{n_t} \Phi(\mathbf{x}_i) \tag{1}$$

and instances are classified by measuring their distance to these prototypes in the feature space. It is an efficient solution for simple class-incremental learning tasks by training only a classifier. However, these methods tend to rely too heavily on pre-trained knowledge and often fail to sufficiently adapt to new classes. This limits their effectiveness in more complex learning scenarios requiring feature-space reorganization.

**Learning Prompts.** This body of works (18; 19; 20; 21) construct and train a learnable pool of prompts that can be shared across all tasks to influence the self-attention process either from the input layer alone or across all layers. This prompt pool with a size of $M$ is denoted as $\mathcal{P} = \{P_1, P_2, \cdots, P_M\}$, where $P_j \in \mathbb{R}^{L_p \times d}$ represents a single prompt with token length $L_p$ and the same embedding size $d$ as image patch embedding $\mathbf{x}_e$. Each prompt is paired with a trainable key vector $k_i \in \mathbb{R}^{d_k}$ encodes task-specific information while preserving the pre-trained backbone $\Phi(\cdot)$, creating a set of key-prompt pairs $\{(k_1, P_1), (k_2, P_2), \cdots, (k_M, P_M)\}$. The training objective jointly optimizes prompts, keys, and classifier through

$$\min_{\mathcal{P}, \mathcal{K}, W} \ell(W^\top \Phi(\mathbf{x}; \mathcal{P}), y) + \lambda \sum_{i=1}^{N} \gamma(\Phi(\mathbf{x}), k_{s_i}) \tag{2}$$

where $\ell(\cdot, \cdot)$ is cross-entropy loss measuring the discrepancy between the prediction and ground truth, $\gamma(\cdot, \cdot)$ measures cosine similarity between keys and queries, $s_i$ denotes the index of the i-th selected key from the key-prompt set, and $\lambda$ balances task performance against prompt selection efficacy. During inference, the model first extracts key features $\Phi(\mathbf{x})$ from the frozen FM without any prompts to solve the prompt retrieval objective

$$K_x^* = \arg \min_{\{s_i\}_{i=1}^N \subseteq [1,M]} \sum_{i=1}^{N} \gamma(\Phi(\mathbf{x}), k_{s_i}), \tag{3}$$

where the *argmin* operator returns the index set $K_x^* = \{s_i\}_{i=1}^N$, which is then used to access the corresponding prompts $\{P_{s_1}, P_{s_2}, \cdots, P_{s_N}\}$. These prompts then condition the transformer's self-attention layers via concatenation with patch embeddings, yielding final predictions through an additional pass on the modified encoder $\Phi(\mathbf{x}; \mathcal{P})$.

Although they present relatively efficient adaptations, selecting the correct prompt for a given task becomes challenging especially in long and complex scenarios, as the fixed key embedding space $\Phi(\cdot)$ struggles to discriminate between semantically similar but task-distinct prompts, leading to retrieval conflicts when $\gamma(k_i, k_j) \approx 1$ for prompts $P_i$, $P_j$ from incompatible tasks, resulting in forgetting.

**Learning Adapters.** These approaches (22; 23; 24) address catastrophic forgetting by inserting lightweight neural modules called adapters into the FM's layers, enabling task-specific adaptations while preserving frozen base parameters. Each set of adapters, $\mathcal{A}_t = \{A_1, A_2, \ldots, A_N\}$, for task $t$ operates via residual connections across $N$ transformer layers. These adapters typically project features through a low-dimensional bottleneck, given an intermediate feature as defined in

$$A(\mathbf{z}) = \mathbf{z} + \psi(\mathbf{z}W_{down})W_{up}, \quad W_{down} \in \mathbb{R}^{d \times r}, \ W_{up} \in \mathbb{R}^{r \times d} \tag{4}$$

where $\mathbf{z}$ represents the output of the MLP block in a transformer layer, $\psi$ denotes a non-linear activation function, typically GELU, and the adapter's projection layers follow the constraint $r \ll d$. Task-specific adapter sets then can be trained using either a feature concatenation strategy or a module merging strategy. Under the feature concatenation strategy (23), adapter sets are trained sequentially for each session and their outputs are concatenated with the FM features at the cost of quadratic scaling or a linear increase in dimensionality. In contrast, the module merging strategy (24) builds on previous adapter sets where each new set $\mathcal{A}_t$ refines the representation produced by the preceding set $\mathcal{A}_{t-1}$ to produce a gradual and unified feature representation. This is more parameter-efficient compared to the feature concatenation strategy, but it risks accumulating feature drift over successive tasks, especially when new class distributions diverge significantly from those of earlier sessions.

Although they modify the pre-trained model's feature representations via residual additions, inserting adapters into all $N$ transformer layers incurs substantial parameter overhead, requiring $(T \times N \times 2dr)$ additional parameters, where $T$ denotes the number of tasks, $r$ is the bottleneck projection dimension, and $d$ is the embedding size. Moreover, these residual modifications introduce subtle yet cumulative deviations from the original pre-trained feature space, which become particularly pronounced in deeper layers. Consequently, while individual adapters are lightweight, their pervasive placement across layers poses challenges for overall parameter efficiency during both training and inference.

## 4 Methodology

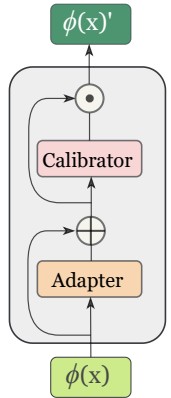

Figure 2: LuCA.

Here, we first introduce our general-purpose PEFT module LuCA and then present the details of TOSCA, a specialized instantiation of LuCA designed for post-training in CIL, with theoretical insights into the stability-plasticity trade-off of the feature manifolds.

**LuCA Module.** Our module is based on an adapter-style design, where it learns refined task-specific information. In particular, LuCA separates feature transformation from discriminative feature enhancement through a dual adapter–calibrator structure, enabling fine-grained control over parameter updates. LuCA can process any intermediate representation $\mathbf{z} \in \mathbb{R}^d$ through two sequential operations:

$$L(\mathbf{z}) = C(A(\mathbf{z})), \tag{5}$$

where $A(\cdot)$ is a residual adapter that applies bottlenecked feature modulation with Eq. (4) to preserve original semantics via skip connections while learning task-specific offsets. The calibrator $C(\cdot)$ then reweights the adapter's output features through an attention-like gating, and refines the more discriminative features with

$$C(\mathbf{z}) = \mathbf{z} \odot \sigma(\mathbf{z}V_{down})V_{up}, \quad V_{down} \in \mathbb{R}^{d \times r}, \ V_{up} \in \mathbb{R}^{r \times d} \tag{6}$$

where $\odot$ denotes the Hadamard product and $\sigma$ indicates a sigmoid activation function. We call this component a 'calibrator' because its gating mechanism performs task-dependent feature calibration: it produces a soft importance mask that rescales each feature dimension according to how informative it is for the current task. This operation corrects over-activated or noisy channels while amplifying stable and discriminative ones, thereby calibrating the representation before it is passed to the classifier. Compared to complete fine-tuning that scales with $\mathcal{O}(d^2)$, LuCA provides an efficient and flexible mechanism for task adaptation with only $4 \times d \times r$ trainable parameters, leading to a significantly reduced $\mathcal{O}(dr)$ complexity, where $r \ll d$.

**TOSCA: Specialization for Continual Post-Training.** In this work, to enable continual post-training with neuroscientific inspirations, we instantiate the LuCA module as TOSCA which is a strategic implementation operating exclusively on the final `[CLS]` token just before the classifier. Given an input $\mathbf{x}$, the frozen pre-trained backbone generates features $\Phi(\mathbf{x})$ of the last `[CLS]` token, and TOSCA refines them through

$$\Phi(\mathbf{x})' = L(\Phi(\mathbf{x})) = C(A(\boldsymbol{\Phi}(\mathbf{x}))). \tag{7}$$

The design of placing it at the last token is a deliberate architectural choice with three advantages: First, by localizing adaptation to the final `[CLS]` token, TOSCA preserves the feature hierarchy: low- and mid-level representations remain stable, while only the high-level abstractions adapt to new tasks. This minimizes disruption to learned invariant features *(stability)* while still injecting flexible task-specific adjustments *(plasticity)* at the final semantic aggregation point, mirroring the functional synergy between the ventral visual stream and the prefrontal cortex where stable representations from the ventral stream (26; 27) are integrated and modulated by task-specific circuits in the prefrontal cortex (28; 29; 30) just before driving behavioral responses. Second, the last `[CLS]` token inherently aggregates all semantic information, making it an optimal locus for task-specific refinement, in contrast to input-layer modifications of prompt-based approaches, which indirectly influence later representations through the self-attention mechanism. Third, this design avoids modifying multiple layers and ensures that the total parameter count remains architecture-agnostic, with a fixed footprint of $4dr$ that does not scale with model depth. In contrast, layer-wise adapters scale linearly as $N \times 2dr$ for $N$ layers. This significant reduction in parameters leads to decreased training and inference complexity.

**Training Protocol.** We completely freeze the parameters of FM and only train the TOSCA's parameters $\Theta = \{W_{down}, W_{up}, V_{down}, V_{up}\}$ together with the prototypical classifier $W^\top$. We utilize a new TOSCA module for each incremental stage $t$ with the parameters $\Theta_t$, which encodes task-specific information by optimizing a composite objective function that combines cross-entropy loss with $\ell_1$-regularization as

$$\min_{\Theta_t \cup W^\top} \sum_{(\mathbf{x},y) \in \mathcal{D}^t} \ell_{CE}\left(W^\top \Phi(\mathbf{x})'_t, y\right) + \lambda \|\Theta_t\|_1, \quad \Phi(\mathbf{x})'_t = L_t(\Phi(\mathbf{x})) \tag{8}$$

where $\lambda$ controls the regularization strength. The $\ell_1$ term induces to use of only a sparse subset of weights in the module, which encourages orthogonality. This orthogonal specialization enables each module to specialize on distinct feature dimensions, preventing interference between the tasks (62; 63). After training, we store $\Theta_t$ while keeping the pre-trained backbone $\Phi(\cdot)$ intact.

**Inference Protocol.** We divide our inference protocol into a two-stage design for computational efficiency with minimal overhead. In the first stage, a single forward pass through the frozen backbone extracts a shared representation $\Phi(\mathbf{x})$ for the input batch, up to the classifier. In the second stage, each TOSCA module independently processes this representation to produce task-specific predictions. This design avoids redundant computation and enables efficient reuse of features across all modules. Each transformed feature $\Phi(\mathbf{x})'_b$ produces a task-specific probability distribution, and the module with the lowest output entropy is selected to make the final prediction over the union of all classes. This leverages the fact that an appropriate task-specific module typically yields lower uncertainty due to its specialized feature calibration and orthogonality, thereby enabling selection of the relevant module *without access to task labels*. The procedure can be formalized as in Eq. (9), where $H(\cdot)$ denotes the Shannon entropy and $\pi_t$ represents task priors, assumed uniform by default.

$$\hat{y} = \arg\min_{y \in \mathcal{Y}} H\left(\sum_{t=1}^{T} \pi_t p_t(y|\mathbf{x})\right), \quad p_t(y|\mathbf{x}) = softmax(W^\top \Phi(\mathbf{x})'_t) \tag{9}$$

**Summary.** Although prompt-based methods are primarily better at stability by modulating the self-attention dynamics within frozen foundation models, adapter-based strategies focus more on promoting plasticity through lightweight residual adaptations that fine-tune task-specific knowledge. Despite the success of both directions in CIL, they often address only one side of the stability–plasticity dilemma. To combine the complementary benefits of both, we introduce a new general-purpose PEFT module LuCA that integrates an adapter with a calibrator to yield more coherent and refined feature representations. Unlike existing continual post-training methods that place modules at every layer, we strategically position a sparse LuCA module to operate solely on the final [CLS] token just before the classifier, which we refer to as TOSCA. This neuro-inspired design encourages each module to specialize orthogonally in its feature subspace, prevents task interference during inference, and efficiently achieves an elegant balance between stability and plasticity, without relying on complex tricks, offering a principled step forward to post-training in CIL. Please refer to Appendix A.1 for the theoretical underpinnings and algorithmic flow.

## 5 Experiments

In this section, we describe our setup and present results on seven benchmarks, comparing our approach with state-of-the-art algorithms. Additionally, we compare against a joint training performance and provide task-wise accuracies to demonstrate the adaptivity capacity of each method. Finally, we share a parameter and run-time analysis, along with an ablation study and offer deeper insights with further discussion.

### 5.1 Experimental Setup

**Datasets.** Since FMs often exhibit substantial knowledge of upstream tasks, we adopt the evaluation framework proposed in (18; 19; 20; 22; 23; 24) to assess their performance across a diverse set of benchmarks. These include CIFAR-100 (64), CUB-200 (65), ImageNet-R (66) ImageNet-A (67), OmniBenchmark (68), and VTAB (69). These datasets encompass both standard CIL benchmarks and out-of-distribution datasets which exhibit significant domain shifts relative to the dataset used for pre-training, e.g. ImageNet (70). Furthermore, to assess true out-of-distribution performance, we use the EuroSAT (71), a dataset with a fundamentally different satellite imagery domain. CIFAR-100 has 100 classes of natural images, each with 500 training images. CUB-200 dataset consists of images from 200 bird classes, with about 30 images per class for training. ImageNet-R includes 24000 for training images from 200 classes with abstract forms. ImageNet-A consists of 200 classes and 7500 training samples that are usually misclassified by ResNet models. Omnibenchmark with 300 classes and VTAB with 50 classes are designed to evaluate the generalization of visual representations. Finally, EuroSAT contains 10 satellite imagery classes, each with roughly 2,000–3,000 samples, totaling 27,000 geo-referenced images. To perform class-incremental learning, we follow (22; 23; 24) and adopt the notation 'B-$m$ Inc-$n$', where $m$ denotes the number of classes in the initial stage and $n$ indicates the number of classes introduced at each incremental stage.

**Comparison Methods.** We select state-of-the-art continual post-training methods for comparison: SimpleCIL (22), RanPAC (15), L2P (18), DualPrompt (19), CODAPrompt (20), HiDePrompt (21), APER (22), EASE (23) and MOS (24). All of them work under the FM-based replay-free CIL setting, *except HiDePrompt and MOS which generate pseudo-replay from class statistics for classifier alignment.* We also include one lower-bound and one upper-bound reference point: 'Finetune' sequentially fine-tunes the FM; and 'joint' trains the model with all classes at the same time. All methods are implemented using the same FM.

**Evaluation Metrics.** We compare the methods with well-recognized continual learning metrics which are based on the accuracy over all stages obtained after last stage, and the accuracy across all stages (35). We denote the Top-1 accuracy after the $t$-th stage as $\mathcal{A}_t$ and use $\mathcal{A}_T$ to represent the performance after the final stage. The average performance across all incremental stages is then measured by $\bar{\mathcal{A}} = \frac{1}{T}\sum_{t=1}^{T} \mathcal{A}_t$.

Table 1: Average accuracy ($\bar{\mathcal{A}}$) and last accuracy ($\mathcal{A}_T$) on six datasets with **ViT-B/16-IN21K**. 'IN-R/A' stands for 'ImageNet-R/A', and 'OmniBench' stands for 'OmniBenchmark.' We report all compared methods with their source code and show the best performance in bold. '–' denotes non-applicability under the released implementation.

| Method | CIFAR B0 Inc5 | | CUB B0 Inc10 | | IN-R B0 Inc20 | | IN-A B0 Inc20 | | OmniBench B0 Inc30 | | VTAB B0 Inc10 | |
|---|---|---|---|---|---|---|---|---|---|---|---|---|
| | $\bar{\mathcal{A}}$ | $\mathcal{A}_T$ | $\bar{\mathcal{A}}$ | $\mathcal{A}_T$ | $\bar{\mathcal{A}}$ | $\mathcal{A}_T$ | $\bar{\mathcal{A}}$ | $\mathcal{A}_T$ | $\bar{\mathcal{A}}$ | $\mathcal{A}_T$ | $\bar{\mathcal{A}}$ | $\mathcal{A}_T$ |
| Joint | – | 96.21 ±1.0 | – | 92.62 ±1.1 | – | 81.92 ±1.4 | – | 67.97 ±1.9 | – | 85.44 ±1.2 | – | 94.96 ±1.2 |
| Finetune | 60.65 ±5.6 | 48.12 ±3.3 | 55.78 ±2.8 | 33.13 ±3.3 | 59.09 ±3.7 | 49.46 ±3.3 | 30.98 ±3.4 | 19.86 ±1.8 | 63.71 ±1.0 | 45.45 ±1.0 | 31.60 ±6.0 | 21.63 ±8.3 |
| SimpleCIL | 86.48 ±0.8 | 81.28 ±0.1 | 91.58 ±1.3 | 86.73 ±0.1 | 61.31 ±0.4 | 54.55 ±0.1 | 58.92 ±1.0 | 48.77 ±0.1 | 79.59 ±1.5 | 73.13 ±0.1 | 90.65 ±1.1 | 84.43 ±0.1 |
| RanPAC | 93.41 ±0.8 | 90.02 ±0.8 | 93.35 ±0.7 | 89.27 ±0.5 | 81.31 ±0.9 | 75.44 ±1.3 | 64.22 ±1.4 | 54.76 ±1.2 | 83.38 ±1.1 | 78.33 ±0.6 | 94.91 ±1.7 | 91.93 ±1.9 |
| L2P | 84.90 ±1.2 | 80.06 ±1.4 | 73.22 ±1.8 | 61.55 ±1.7 | 75.92 ±0.7 | 70.88 ±0.7 | 50.13 ±1.8 | 42.80 ±1.1 | 73.96 ±2.0 | 64.63 ±0.6 | 78.61 ±4.2 | 64.81 ±2.9 |
| DualPrompt | 85.61 ±1.3 | 79.92 ±0.4 | 81.36 ±1.8 | 70.51 ±1.1 | 71.48 ±0.5 | 66.09 ±1.3 | 51.57 ±0.4 | 40.56 ±1.6 | 75.58 ±1.4 | 66.46 ±0.8 | 86.86 ±2.8 | 75.86 ±3.7 |
| CODAPrompt | 87.64 ±0.4 | 81.46 ±0.3 | 77.65 ±1.0 | 68.44 ±1.0 | 76.25 ±0.3 | 71.39 ±0.3 | 58.82 ±0.78 | 47.18 ±0.9 | 73.73 ±0.5 | 69.46 ±0.7 | 87.60 ±0.5 | 86.71 ±0.8 |
| HiDePrompt | 85.99 ±0.4 | 82.95 ±0.7 | 83.86 ±0.7 | 79.56 ±1.0 | 79.63 ±0.8 | 74.06 ±1.3 | - | - | - | - | - | - |
| APER-Adapter | 89.57 ±0.9 | 84.91 ±0.2 | 91.62 ±1.2 | 86.72 ±0.2 | 74.81 ±0.8 | 66.97 ±0.8 | 59.57 ±1.6 | 49.46 ±0.4 | 80.48 ±1.2 | 74.04 ±0.3 | 90.59 ±1.0 | 84.28 ±0.2 |
| EASE | 90.79 ±0.8 | 85.97 ±0.6 | 92.51 ±1.3 | 86.49 ±1.2 | 80.35 ±1.0 | 75.74 ±0.8 | 64.00 ±1.5 | 54.99 ±1.0 | 81.11 ±0.8 | 74.16 ±2.0 | 90.26 ±3.6 | 82.07 ±3.0 |
| MOS | 93.45 ±0.9 | 90.04 ±0.6 | 93.42 ±1.2 | 90.07 ±0.9 | 82.26 ±1.0 | 77.62 ±0.9 | 63.57 ±2.0 | 54.60 ±0.8 | 84.73 ±1.1 | 79.97 ±0.9 | 92.75 ±1.0 | 92.74 ±0.9 |
| TOSCA (ours) | **96.37** ±0.5 | **95.64** ±0.8 | **93.47** ±1.9 | **91.09** ±1.8 | **82.27** ±1.9 | **79.28** ±1.9 | **66.92** ±3.0 | **65.37** ±2.9 | **84.75** ±2.6 | **82.35** ±1.0 | **96.59** ±1.6 | **93.87** ±2.0 |

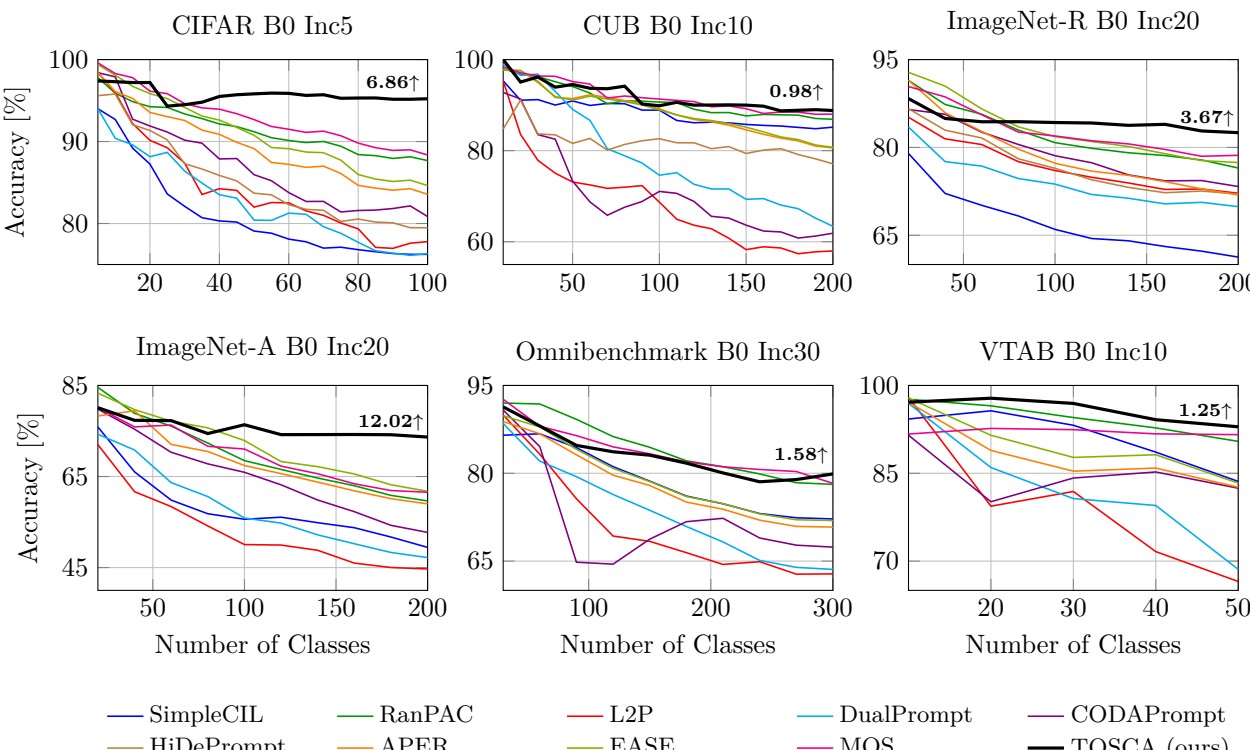

Figure 3: Last accuracy ($\mathcal{A}_T$) after each learning session on different methods under different settings. All methods are initialized with **ViT-B/16-IN1K**. We annotate the relative improvement of TOSCA above the runner-up method with numbers at the last incremental stage.

**Implementation Details.** We conduct experiments on an NVIDIA A100, and reproduce the compared methods using PyTorch (72) and Pilot (73). Consistent with (22; 23; 24), we utilize two representative FMs: ViT-B/16-IN21K and ViT-B/16-IN1K. Both models are pre-trained on ImageNet21K, with the latter further fine-tuned on ImageNet1K. For TOSCA, we train the model with SGD using a batch size of 48 over 20 epochs. The learning rate follows cosine annealing schedule, starting at $2.5e^{-2}$. The projection dimension $r$ is set to 48 and the $\ell_1$ contribution $\lambda$ is set to $5e^{-4}$. We perform multiple runs with five different random seeds and report mean and standard deviation for each method.

Table 2: Task-wise accuracy ($\mathcal{A}_t$) on fine-grained CUB B0 Inc10 with **ViT-B/16-IN21K**. TOSCA returns the highest average task-wise accuracy, illustrating its superior plasticity against existing approaches.

| Task | T1 | T2 | T3 | T4 | T5 | T6 | T7 | T8 | T9 | T10 | T11 | T12 | T13 | T14 | T15 | T16 | T17 | T18 | T19 | T20 | **Avg** |
|---|---|---|---|---|---|---|---|---|---|---|---|---|---|---|---|---|---|---|---|---|---|
| SimpleCIL | 100 | 97.6 | 97.9 | 97.2 | 98.1 | 89.1 | 89.6 | 96.4 | 93.9 | 87.5 | 92.1 | 83.2 | 86.9 | 88.9 | 84.9 | 95.1 | 87.3 | 93.2 | 82.5 | 82.6 | 91.2 |
| RanPAC | 100 | 96.1 | 97.9 | 97.2 | 98.1 | 90.9 | 92.4 | 97.1 | 94.7 | 91.1 | 94.7 | 87.6 | 88.7 | 91.5 | 87.3 | 96.7 | 91.0 | 95.2 | 83.3 | 87.6 | 92.9 |
| L2P | 100 | 85.2 | 71.1 | 77.3 | 85.9 | 91.8 | 71.7 | 87.2 | 78.2 | 64.7 | 85.2 | 59.3 | 79.4 | 66.1 | 62.7 | 75.8 | 70.2 | 59.2 | 54.5 | 59.2 | 74.2 |
| DualPrompt | 100 | 96.9 | 97.4 | 90.9 | 95.3 | 81.9 | 76.4 | 95.0 | 88.7 | 77.2 | 86.1 | 73.4 | 84.1 | 58.5 | 59.2 | 75.8 | 76.8 | 69.9 | 62.1 | 58.2 | 80.2 |
| CODAPrompt | 97.3 | 98.3 | 72.4 | 87.4 | 97.6 | 85.7 | 59.6 | 84.4 | 93.9 | 89.3 | 80.8 | 69.1 | 59.8 | 83.8 | 46.4 | 40.9 | 72.9 | 61.9 | 94.3 | 86.9 | 78.1 |
| HiDePrompt | 99.5 | 92.6 | 77.2 | 86.2 | 82.9 | 90.9 | 70.3 | 90.8 | 92.9 | 92.3 | 88.5 | 79.1 | 67.8 | 87.0 | 65.7 | 88.5 | 87.5 | 80.7 | 90.6 | 88.6 | 84.9 |
| APER-Adapter | 100 | 97.6 | 97.9 | 97.3 | 98.1 | 89.1 | 89.6 | 96.4 | 94.7 | 87.5 | 92.2 | 83.2 | 86.9 | 88.9 | 84.9 | 95.1 | 87.3 | 93.2 | 82.6 | 82.6 | 91.3 |
| EASE | 100 | 97.6 | 97.9 | 97.2 | 97.2 | 88.3 | 93.4 | 96.5 | 95.5 | 88.2 | 92.2 | 87.6 | 87.8 | 88.9 | 86.5 | 94.3 | 88.1 | 95.2 | 85.6 | 86.7 | 92.2 |
| MOS | 98.3 | 97.7 | 97.9 | 98.1 | 98.1 | 95.5 | 94.3 | 97.8 | 94.7 | 91.1 | 95.6 | 91.1 | 89.0 | 93.2 | 89.6 | 95.9 | 91.0 | 90.9 | 88.5 | 92.8 | 94.1 |
| TOSCA (ours) | 100 | 97.7 | 98.9 | 99.1 | 100 | 96.4 | 100 | 100 | 100 | 97.1 | 99.1 | 99.1 | 90.6 | 91.5 | 99.3 | 96.8 | 99.2 | 99.0 | 94.7 | 93.9 | **97.6** |

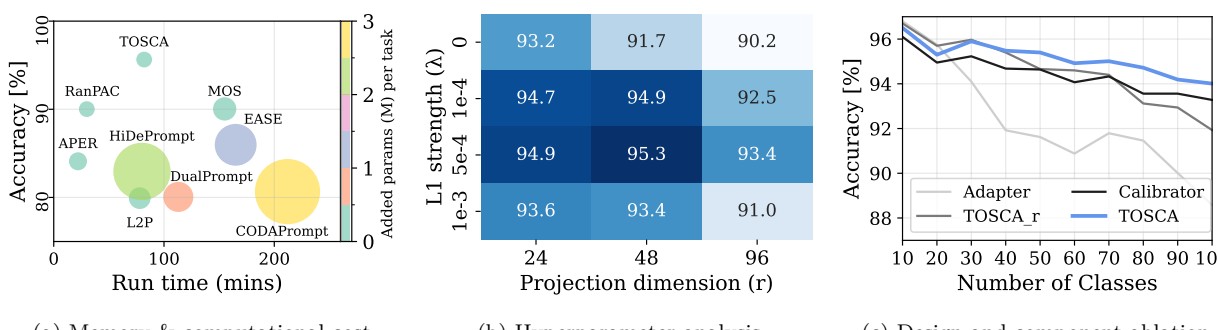

(a) Memory & computational cost  (b) Hyperparameter analysis  (c) Design and component ablation

Figure 4: Performance of TOSCA across different perspectives. (a) Memory & computational cost highlights TOSCA's efficiency, (b) Hyperparameter analysis illustrates effect of $\ell_1$ strength ($\lambda$) and projection dimension ($r$) on accuracy, (c) Design and component ablation presents the impact of different components and flows on accuracy.

## 5.2 State-of-the-Art Comparison

We compare TOSCA with leading state-of-the-art continual learning methods across six benchmark datasets and multiple pre-trained foundation models. Table 1 summarizes the final-stage accuracy using ViT-B/16 pre-trained on IN21K for each method. TOSCA consistently achieves the highest performance across all six benchmarks, significantly surpassing existing approaches such as EASE and MOS. Notably, MOS relies on replay generation for classifier alignment and is therefore not strictly replay-free, whereas TOSCA achieves superior results without requiring a replay. To further examine learning dynamics, we report the incremental performance trends over successive training sessions in Figure 3. Across datasets, TOSCA outperforms the next-best methods by $4\%-12\%$ on CIFAR100, ImageNet-R, and ImageNet-A, as indicated by the annotations at the end of learning sessions. These improvements are particularly pronounced on out-of-distribution datasets, highlighting TOSCA's robustness and capacity to generalize under distributional shifts. Please see our Appendix A.2 for more results.

## 5.3 Comparison to Joint Training

To provide a more complete picture, we also compare TOSCA to a traditional 100-epoch joint training baseline, which represents an upper-bound for performance since it has access to all data at once. While joint training still holds the top spot, TOSCA demonstrates highly competitive performance despite only training a single, lightweight module for each new task. This is a crucial finding, as TOSCA delivers near-optimal performance while drastically reducing computational cost and training time. Unlike full joint training, it avoids the need to revisit all past data, making it a practical and scalable solution for real-world continual learning scenarios where memory and compute are limited.

Table 3: Task-wise accuracy ($\mathcal{A}_t$) on out-of-distribution scenario with EuroSAT B0 Inc2 to show adaptation of each approach.

| Task | T1 | T2 | T3 | T4 | T5 | **Avg** |
|------|------|------|------|------|------|------|
| SimpleCIL | 99.7 | 85.1 | 85.8 | 70.0 | 88.8 | 85.9 |
| RanPAC | 100 | 98.6 | 98.2 | 95.2 | 98.8 | 98.1 |
| L2P | 99.9 | 89.6 | 56.2 | 46.0 | 91.7 | 76.7 |
| DualPrompt | 100 | 92.6 | 60.6 | 23.7 | 75.5 | 70.5 |
| CODAPrompt | 99.9 | 88.8 | 92.8 | 90.0 | 72.1 | 88.7 |
| APER-Adapter | 99.9 | 87.2 | 86.2 | 67.3 | 90.8 | 86.3 |
| EASE | 99.9 | 93.6 | 92.1 | 71.1 | 93.4 | 90.0 |
| MOS | 100 | 96.1 | 96.0 | 93.6 | 98.2 | 96.8 |
| TOSCA (ours) | 99.9 | 99.6 | 99.3 | 98.8 | 98.8 | **99.3** |

Figure 5: Last accuracy ($\mathcal{A}_T$) performance after each learning session on EuroSAT B0 Inc2.

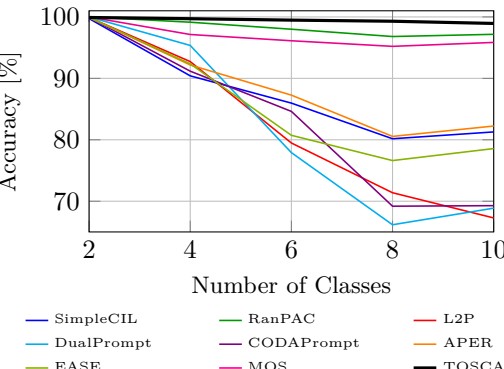

## 5.4 Task-Wise Plasticity

Plasticity is a critical property for continual learning models, as it reflects the model's ability to effectively adapt to new tasks. A potential concern for the plasticity of TOSCA may be that its adaptation is restricted to a single module that acts only on the final layer's `[CLS]` token, whereas other methods perform adjustments across multiple layers. To investigate this, we give a detailed breakdown of the task-wise accuracy performance, evaluating across all methods. In particular, to truly evaluate plasticity, we examine performance on the CUB B0 Inc10 scenario which is highly fine-grained and presents a long sequence of tasks, and stresses the model's ability to distinguish subtle inter-class differences over many incremental learning stages. Our results in Table 2 show that TOSCA consistently achieves high task-wise accuracy, often matching or surpassing methods that adapt multiple layers. This indicates that, despite its single-module design, TOSCA is capable of strong task-specific adaptation. We attribute this performance to the strategic placement of the module just before the classification layer, which allows it to modulate features directly relevant to the decision without disrupting the pre-trained, stable representations in earlier layers.

## 5.5 Parameter and Run-Time Analysis

We further investigate FM-based continual post-training approaches in terms of accuracy, computational cost, and parameter efficiency on CIFAR B0 Inc5 benchmark in Figure 4a. TOSCA achieves the top performance while maintaining a low computational cost and parameter overhead per task. In contrast, methods like CODAPrompt and EASE require significantly more parameters and longer run times, making them less efficient. Notably, MOS also attains high accuracy, but it comes at a higher computational expense due to additional processes such as adapter merging and replay generation. Overall, TOSCA demonstrates its effectiveness in CIL with minimal parameter overhead and shorter run-time, striking a balance between efficiency and performance.

## 5.6 Out-of-Distribution Performance

Although existing continual learning benchmarks incorporate certain levels of domain shift, evaluating robustness under more drastic distribution changes remains an open and compelling testbed. We therefore compare all methods on the EuroSAT benchmark which introduces a substantial domain shift to assess true out-of-distribution performance. Our results in Figure 5 reveal that all approaches except RanPAC, MOS and TOSCA exhibit substantial performance degradation as tasks progress. While some methods achieve strong task-wise performance, they fail to retain this knowledge incrementally. For example, EASE attains high per-task accuracy with an average of 90% in Table 3, yet its overall incremental performance drops to 78.5%, highlighting pronounced forgetting under distribution shift. Despite the significant domain shift, TOSCA outperforms all competing approaches, demonstrating more robust generalization beyond the pre-training distribution. Notably, this performance gain is achieved with minimal additional computation, reinforcing TOSCA as a practical and scalable solution for continual learning under severe distribution shifts.

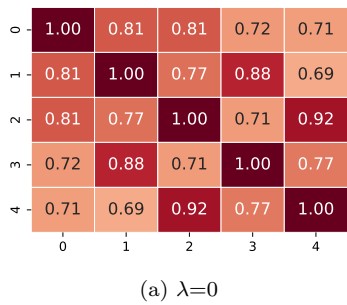 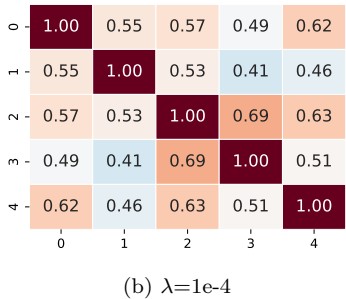 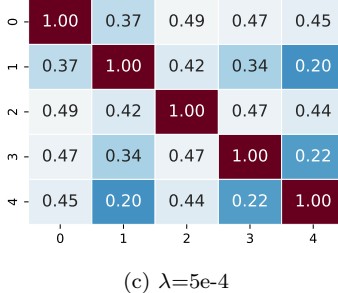

(a) $\lambda$=0         (b) $\lambda$=1e-4         (c) $\lambda$=5e-4

Figure 6: Cosine similarity of modules with increasing $\ell_1$ strength ($\lambda$). It encourages modules to become more orthogonal, leading to learn different representations while facilitating to select the most suitable module at inference.

## 5.7 Ablation Study

We also perform an ablation study on CIFAR B0 Inc10, evaluating the incremental performance across different learning settings. First, we analyze the impact of $\ell_1$-regularization strength ($\lambda$) and projection dimension ($r$) on performance, as shown in Figure 4b. Our findings indicate that $\ell_1$ regularization improves accuracy, with performance reaching peak at $\lambda = 5e^{-4}$. This promotes orthogonality among different modules, improving module selection during inference. However, excessive sparsity degrades performance by excessively constraining representations, thereby reducing expressiveness and learning capacity. Similarly, increasing the projection dimension ($r$) improves accuracy up to $r = 48$, beyond which performance deteriorates due to the larger bottleneck. Based on these observations, we identify the optimal configuration as $\lambda = 5e^{-4}$ and $r = 48$, achieving an accuracy of 95.3%. Additionally, we compare the performance of different components, alternative module designs and configurations against TOSCA in Figure 4c. This includes a reversed variant, TOSCA_r, which integrates new information atop the calibrated pre-trained features, formulated as $\Phi(\mathbf{x})' = A(C(\Phi(\mathbf{x})))$. We attribute this behavior to the ordering of the modules: when calibration operates on unmodified features before adaptation, the flexibility to modify representations is reduced, whereas introducing task-specific adjustments first and selectively regularizing them after calibration enables more effective adaptation on newly acquired features, explaining the performance gap between TOSCA and TOSCA_r. Therefore, our results highlight the crucial role of the calibrator while TOSCA surpasses all variants by effectively harmonizing its two modules working together. We further discuss this behavior and provide a step-by-step analysis of the functioning of our approach below.

## 5.8 Further Analysis and Discussion

We first analyze the effect of the $\ell_1$ on the learned module representations by computing pairwise cosine similarities between modules under different regularization strengths. We reveal in Figure 6 that a monotonic drop in off-diagonal cosine similarity as $\lambda$ increases, indicating substantial redundancy and overlap in their functional roles. The underlying mechanism is straightforward: $\ell_1$ regularization induces sparsity, and sparsity reduces overlap in the active feature dimensions across modules. When two modules rely on fewer shared coordinates, their output directions exhibit lower cosine similarity. In other words, $\ell_1$ pushes modules toward using disjoint subsets of features, which manifests as reduced correlation and hence more 'orthogonal' representations in practice. This explanation aligns precisely with the trends in Figure 4b: at $\lambda = 5e^{-4}$, the representations become substantially more decorrelated, with similarities dropping toward $\approx 0.3$–0.4.

We next analyze how architectural components influence representational structure using t-SNE embeddings of `[CLS]` features under three configurations as presented in Figure 7 on the first learning session of CUB B0 Inc10 with ViT-B/16-IN21K. The pre-trained ViT exhibits diffuse and overlapping cluster geometry, producing an inter/intra-class distance ratio of 1.97. Adding the adapter expands and separates clusters by performing task-sensitive residual transformations while geometrically shifting representations towards a new task-relevant subspace, yielding a class-separation measure of 5.32, around 1.5× improvement over the pre-trained baseline, but clusters remain elongated and partially entangled. Incorporating the calibrator after the adapter leads to compact, well-aligned, and cleanly separated clusters, achieving a class-separation distance of 9.19, corresponding to an almost 4× improvement.

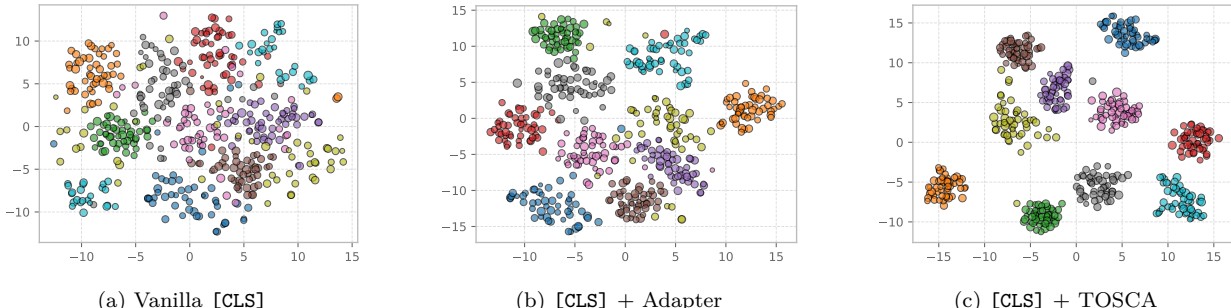

(a) Vanilla `[CLS]`        (b) `[CLS]` + Adapter        (c) `[CLS]` + TOSCA

Figure 7: t-SNE visualization of different architectural choices after the first learning session on CUB B0 Inc10. (a) Vanilla pretrained ViT shows substantial overlap in the `[CLS]` token space. (b) Adding an adapter improves separability but still exhibits noticeable spread and mixing. (c) TOSCA module further calibrates representations, leading to a clearly distinct clusters with minimal overlap.

This refinement occurs because the calibrator amplifies or suppresses the adapter's task-specific signal according to relevance, which in result creates shrunk and well-grouped clusters. Critically, this adapter-then-calibrator ordering ensures that first the adjustment is introduced, then selectively regularized; inverting this order weakens enforcement, as calibration would operate on unmodified features before task-specific adaptation occurs.

Together, these analyses clarify the behavior of our approach: $\ell_1$ regularization encourages modules to rely on distinct subsets of representation space, yielding decorrelated outputs that function as orthogonal experts. Meanwhile, the adapter and calibrator produce complementary geometric effects by separating and then refining class structure and ultimately driving the gains observed in our ablations and main results.

## 6 Conclusions

In this paper, we introduced LuCA, a new parameter-efficient fine-tuning module that combines a lightweight adapter with a calibrator to refine feature representations and facilitate the integration of new information. Building upon this, we proposed TOSCA, a neuro-inspired continual post-training framework based on foundation models, which leverages a single sparse LuCA module applied exclusively to the final `[CLS]` token before the classifier for each task. This targeted adaptation enables highly efficient, task-specific modulation while maintaining orthogonality between tasks, resulting in minimal memory and computational overhead. Extensive experiments across multiple benchmark datasets demonstrate that TOSCA consistently achieves state-of-the-art performance. It effectively balances the stability–plasticity trade-off, outperforming prior methods with substantially fewer additional parameters, while retaining a scalable and model-agnostic design suitable for advancing continual learning with foundation models.

**Limitations and future works.** Although TOSCA shows strong performance, it relies on pre-trained foundation models and its effectiveness depends on the generalizability of these models when adapting through a single token. Future work would explore extending TOSCA to additional modalities or multimodal models and diverse continual learning scenarios, including few-shot, blurry, and class-revisiting incremental settings.

## Broader Impact

This paper aims to advance the field of machine learning, with a focus on replay-free class-incremental post-training. By introducing a lightweight, trainable module that enables continual adaptation without the need to store past data, the proposed approach addresses key concerns related to privacy, memory efficiency, computational overhead, and scalability. These properties make it particularly suitable for deployment in privacy-sensitive or resource-constrained settings, such as personalized models in healthcare, adaptive vision systems for assistive technologies. While the method does not raise immediate ethical concerns, its application in high-stakes domains may require additional safeguards to ensure responsible use.

## Acknowledgements

This work is supported by the EU Horizon programme through SYNERGIES, a project under GA No. 101146542; and ELLIOT, a project under GA No. 101214398; and Dutch e-infrastructure with the support of SURF Cooperative using GA no. EINF-10242; and Turkish MoNE scholarship.

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

# A  Appendix

## A.1  Compared Methods and TOSCA

Here, we provide an overview of the methods evaluated in the main paper. To ensure a fair and consistent basis for comparison, all methods utilize the same FM. This standardization allows us to isolate the contributions of each method's unique approach and compare their performance more accurately. Additionally, we present the pseudocode for TOSCA, providing a clear and detailed description of its working algorithm. This helps to better understand how TOSCA operates, offering insights into its efficiency and functionality within the context of class-incremental post-training.

**Joint**: This method adheres to the traditional supervised batch learning paradigm, where all classes are presented simultaneously and trained over multiple epochs. It serves as the upper bound for class-incremental learning methods, as it does not experience forgetting.

**Finetune**: This method updates all parameters of the pre-trained model when continually trained on new tasks. While it can achieve strong performance, it is susceptible to catastrophic forgetting, where previous knowledge is lost when learning new tasks.

**SimpleCIL** (22): SimpleCIL uses the FM in its original form, combined with a prototypical classifier. It constructs a prototype for each class and utilizes a cosine classifier for classification, aiming for efficient task learning without additional adaptations.

**RanPAC** (15) extends SimpleCIL by randomly projecting the features into the high-dimensional space and learning the analytical ridge classifier in the Gram space for final classification.

**L2P** (18): L2P integrates visual prompt tuning into class-incremental learning with a pre-trained Vision Transformer (ViT). The method places the prompt only in the initial embedding layer, ensuring that the prompt adjusts the features at the early stage of the model while maintaining the frozen structure of the rest of the pre-trained model.

**DualPrompt** (19): DualPrompt builds on L2P by introducing two types of prompts: general prompts (G-Prompt) and expert prompts (E-Prompt). The G-Prompts are applied to the earlier transformer blocks, allowing for broad task-specific adaptation. E-Prompts, on the other hand, are used in the latter blocks of the transformer, providing more specialized tuning for later stages of task processing. This separation allows for more efficient adaptation.

**CODAPrompt** (20): It addresses the challenges of selecting instance-specific prompts by introducing prompt reweighting. It enhances the selection process through an attention mechanism that dynamically weights prompts, improving task-specific performance.

**HiDePrompt** (21): This method leverages task-specific prompts and class-level feature statistics to perform implicit feature-space replay, explicitly optimizing task-identity and task-adaptive prediction. While effective for classifier calibration under self-supervised pre-training, it introduces additional optimization stages and bookkeeping of representation statistics.

**APER** (22): This approach builds on SimpleCIL by introducing an adapter to each transformer layer, but only for the initial task. This adapter helps the pre-trained model to extract task-specific features during the first incremental phase, ensuring better adaptation to the new task while minimizing forgetting in subsequent tasks.

**EASE** (23): This method adds adapters to each layer of FM for every task. This approach leads to good performance by concatenating the feature representations of multiple task-specific backbones, but it comes with an increase in model complexity due to the addition of task-specific adapters at every stage.

**MOS** (24): It also trains adapters for each FM layer for every task. However, MOS introduces the concept of adapter merging and replay generation for classifier correction. These processes increases computational complexity, particularly during training, as the model must handle the merging of multiple task-specific adapters with an increasing number of parameters.

---

**Algorithm 1 TOSCA for Continual Post-Training**

---

**Require:** Incremental datasets: $\{\mathcal{D}^1, \mathcal{D}^2, \ldots, \mathcal{D}^t\}$,
         Pre-trained embedding: $\Phi(\mathbf{x})$

1: **for** $t = 1, 2, \ldots, T$ **do**
2:      Get the incremental training set $\mathcal{D}^t$
3:      Initialize a module $L_t$ with parameters $\Theta_t$ on top of last `[CLS]` token
4:      Optimize the parameters $\Theta_t$ of the module $L_t$ and prototypical classifier $W^\top$ via Eq. (8)
5:      Test the model with all classes seen so far via Eq. (9)
6: **end for**

---

**TOSCA (ours)**: It trains a lightweight adapter–calibrator pair per task on the final `[CLS]` token immediately before classifier. This neuroscience-inspired formulation reduces computational and memory overhead without sacrificing expressiveness, leading to better-calibrated representations and improved performance. Please see our Algorithm 1 for formal and more detailed flow.

**Theoretical underpinnings of our approach.** The core theoretical strength of TOSCA lies in its ability to preserve these representations by avoiding broad modifications to the feature space and instead localizing adaptation to the final layer's `[CLS]` token, while existing methods alter the feature space throughout the network and risk compromising the high-quality representations carefully learned from FMs. Its surgical design not only safeguards representational fidelity, but also provides a principled mechanism to maintain stability while enabling precise, task-specific plasticity. Specifically, the design of TOSCA guarantees that, for all layers $n < N$, the feature manifolds $\mathcal{H}_n$ of the FM are preserved as in Eq. (10), meaning the feature distributions remain identical to the FM's distributions up to the penultimate layer:

$$\forall n < N : \mathcal{H}_n^{TOSCA} = \mathcal{H}_n^{PTM} \tag{10}$$

Adapting solely through the `[CLS]` token of the final layer $N$, TOSCA allows a small bounded deviation in the feature manifold while maintaining backward compatibility of the pre-trained features. This bounded deviation mechanism can be expressed as Eq. (11), where $\epsilon$ is a small constant controlled by the residual connection and constrains changes in the feature space, ensuring that the representations remain robust while providing task-specific flexibility during post-traning, thus achieving a principled stability–plasticity balance.

$$\gamma(\mathcal{H}_N^{PTM}, \mathcal{H}_N^{TOSCA}) \leq \epsilon \tag{11}$$

## A.2 Additional Results

In this section, we provide Table A for pre-trained **ViT-B/16-IN1K** and Figure A that illustrate each incremental step with **ViT-B/16-IN21K**, showcasing the performance of our proposed method. The slight drop in average accuracy $\bar{\mathcal{A}}$ observed in few benchmarks with ViT-B/16-IN1K, compared to existing approaches, stems from its relatively limited generalization capacity compared to ViT-B/16-IN21K, given that TOSCA introduces only a single trainable module per task. This design choice is intentional, prioritizing computational and parameter efficiency while maintaining adaptability in continual learning settings. Notably, our approach consistently achieves superior last incremental accuracy $\mathcal{A}_t$ across all evaluated datasets, underscoring its effectiveness. Overall, TOSCA improves the incremental performance with a minimal overhead cost during both the training and inference phases, emphasizing the efficiency of our method, and making it highly suitable for real-world applications. Furthermore, to verify that TOSCA accurately selects the appropriate adapter–calibrator pair at inference time, we evaluate the accuracy of its entropy-based module selection strategy across all incremental tasks on CUB B0 Inc10. Specifically, we report the percentage of samples for which the selected module matches the ground-truth task identity. The results, summarized in Table B, demonstrate the high reliability of the proposed selection mechanism, especially with higher $\ell_1$ strength.

Table A: Average accuracy ($\bar{\mathcal{A}}$) and last accuracy ($\mathcal{A}_T$) on six datasets with **ViT-B/16-IN1K**. 'IN-R/A' stands for 'ImageNet-R/A,' and 'OmniBench' stands for 'OmniBenchmark.' We report all compared methods with their source code and show the best performance in bold. '–' denotes non-applicability under the released implementation.

| Method | CIFAR B0 Inc5 | | CUB B0 Inc10 | | IN-R B0 Inc20 | | IN-A B0 Inc20 | | OmniBench B0 Inc30 | | VTAB B0 Inc10 | |
| | $\bar{\mathcal{A}}$ | $\mathcal{A}_T$ | $\bar{\mathcal{A}}$ | $\mathcal{A}_T$ | $\bar{\mathcal{A}}$ | $\mathcal{A}_T$ | $\bar{\mathcal{A}}$ | $\mathcal{A}_T$ | $\bar{\mathcal{A}}$ | $\mathcal{A}_T$ | $\bar{\mathcal{A}}$ | $\mathcal{A}_T$ |
|---|---|---|---|---|---|---|---|---|---|---|---|---|
| Joint | – | 95.88 ±1.0 | – | 90.19 ±1.4 | – | 83.87 ±1.4 | – | 74.05 ±1.9 | – | 83.08 ±1.1 | – | 93.24 ±1.8 |
| Finetune | 44.4 ±8.4 | 39.7 ±6.1 | 57.27 ±2.9 | 34.76 ±1.1 | 66.96 ±3.2 | 53.64 ±1.5 | 28.64 ±4.5 | 14.26 ±1.8 | 63.35 ±2.1 | 45.70 ±1.0 | 67.84 ±4.9 | 51.12 ±5.6 |
| SimpleCIL | 82.21 ±0.7 | 76.24 ±0.1 | 90.42 ±1.4 | 85.16 ±0.1 | 66.89 ±0.5 | 61.27 ±0.1 | 58.70 ±1.1 | 49.44 ±0.1 | 78.67 ±1.4 | 72.20 ±0.1 | 90.50 ±1.2 | 83.61 ±0.1 |
| RanPAC | 92.43 ±0.4 | 87.92 ±0.8 | 91.05 ±0.8 | 86.77 ±1.1 | 83.11 ±1.3 | 76.59 ±0.6 | 69.60 ±1.1 | 59.64 ±0.7 | 83.78 ±0.8 | 77.80 ±0.3 | 92.26 ±1.6 | 90.54 ±1.4 |
| L2P | 83.37 ±1.7 | 78.64 ±1.6 | 70.64 ±1.7 | 58.70 ±1.1 | 77.22 ±0.5 | 72.35 ±0.3 | 52.32 ±2.2 | 44.30 ±0.8 | 72.76 ±1.8 | 63.10 ±0.6 | 81.25 ±3.0 | 66.71 ±1.7 |
| DualPrompt | 82.41 ±1.7 | 76.39 ±0.6 | 75.78 ±2.2 | 63.47 ±1.5 | 74.37 ±0.5 | 69.58 ±2.0 | 56.42 ±1.1 | 46.99 ±0.3 | 73.21 ±1.8 | 63.63 ±0.8 | 82.84 ±4.7 | 70.39 ±5.5 |
| CODAPrompt | 86.67 ±0.5 | 80.68 ±1.1 | 70.75 ±1.1 | 61.61 ±1.1 | 78.37 ±0.5 | 73.07 ±0.5 | 63.61 ±0.9 | 52.32 ±0.4 | 72.22 ±0.3 | 68.26 ±0.6 | 84.88 ±1.1 | 82.94 ±1.6 |
| HiDePrompt | 84.67 ±1.5 | 79.59 ±1.3 | 81.46 ±1.6 | 77.11 ±0.9 | 80.32 ±1.5 | 76.06 ±0.4 | - | - | - | - | - | - |
| APER-Adapter | 88.46 ±0.8 | 83.16 ±0.4 | 87.64 ±1.2 | 80.63 ±0.1 | 78.25 ±0.5 | 72.07 ±0.8 | 66.86 ±1.3 | 58.83 ±0.2 | 77.66 ±1.0 | 70.72 ±0.4 | 89.59 ±1.2 | 82.60 ±0.1 |
| EASE | 89.94 ±1.0 | 84.39 ±0.6 | 87.93 ±1.2 | 81.00 ±0.3 | 82.96 ±0.3 | 77.45 ±0.1 | 70.49 ±1.6 | 62.36 ±0.5 | 78.40 ±0.8 | 71.60 ±1.0 | 90.71 ±1.6 | 83.39 ±0.7 |
| MOS | 92.71 ±1.1 | 88.82 ±0.7 | **92.24** ±0.9 | 88.02 ±0.2 | 83.53 ±0.7 | 78.94 ±0.3 | 69.14 ±1.1 | 61.24 ±1.8 | **85.33** ±1.1 | 78.28 ±0.5 | 91.81 ±0.5 | 91.77 ±0.2 |
| TOSCA (ours) | **96.03** ±0.9 | **95.37** ±0.7 | 91.55 ±1.8 | **89.05** ±1.9 | **83.57** ±0.6 | **82.25** ±0.6 | **74.48** ±2.1 | **72.30** ±1.8 | 82.48 ±1.8 | **79.65** ±1.2 | **94.33** ±2.0 | **91.80** ±1.9 |

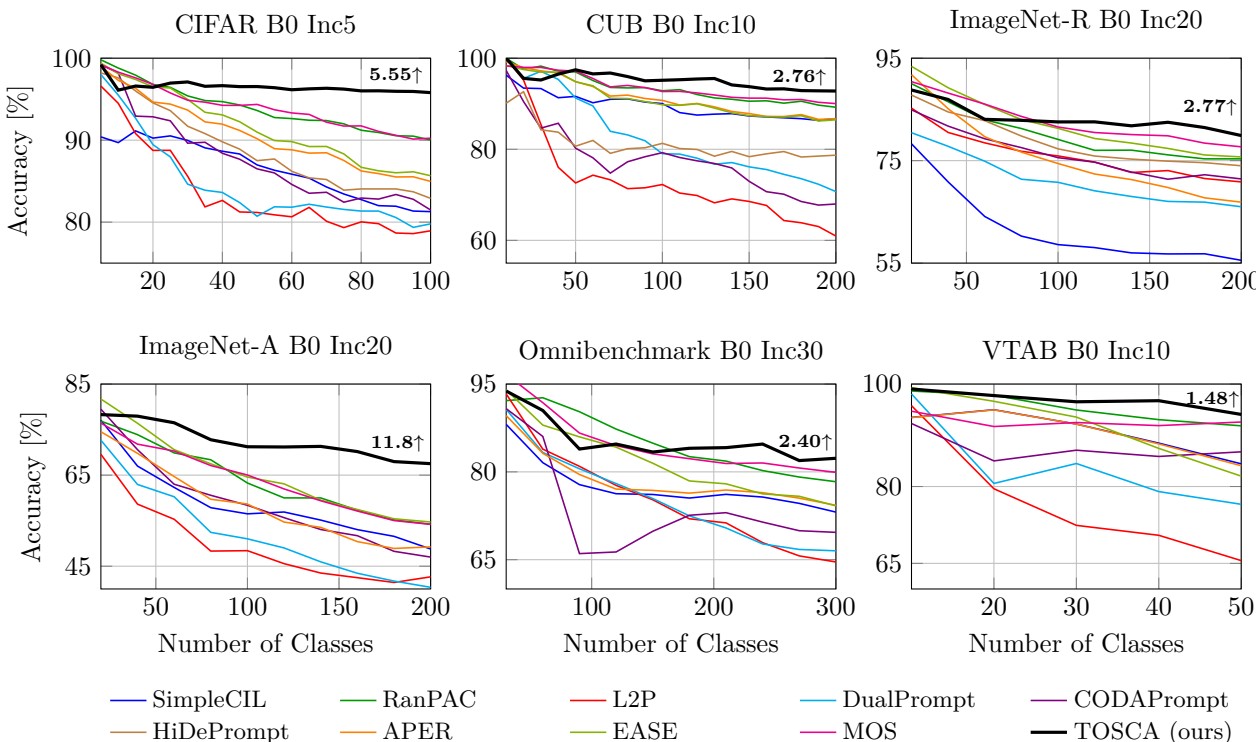

Figure A: Last accuracy ($\mathcal{A}_T$) after each learning session under different settings with **ViT-B/16-IN21K**. Relative improvement of TOSCA is annotated above the runner-up method at the last incremental stage.

Table B: Per-task module selection success rate [%] on CUB B0 Inc10 across different $\ell_1$ regularization strengths. Increased strength leads to more distinct modules and better module selection performance.

| Task | T1 | T2 | T3 | T4 | T5 | T6 | T7 | T8 | T9 | T10 | T11 | T12 | T13 | T14 | T15 | T16 | T17 | T18 | T19 | T20 |
|---|---|---|---|---|---|---|---|---|---|---|---|---|---|---|---|---|---|---|---|---|
| $\lambda = 0$ | 88.2 | 85.4 | 83.9 | 87.1 | 84.3 | 82.6 | 92.5 | 94.1 | 86.7 | 93.2 | 84.9 | 88.7 | 90.2 | 83.4 | 89.3 | 87.1 | 91.5 | 85.6 | 88.3 | 92.1 |
| $\lambda = 1e-4$ | 92.4 | 89.7 | 90.1 | 92.6 | 90.8 | 88.9 | 97.3 | 98.1 | 91.5 | 98.7 | 89.6 | 92.9 | 94.3 | 88.7 | 93.5 | 91.2 | 95.4 | 90.6 | 92.9 | 96.1 |
| $\lambda = 5e-4$ | 95.3 | 94.1 | 96.3 | 94.2 | 92.9 | 92.5 | 100 | 100 | 93.4 | 100 | 94.7 | 96.5 | 97.8 | 100 | 96.9 | 95.1 | 98.6 | 94.3 | 96.8 | 99.2 |

