# OpenReview forum: "Unlocking [CLS] Features for Continual Post-Training"
_TMLR — Accepted by TMLR_

### Review · Reviewer_x8Rr · 2025-12-05

**Summary Of Contributions:**

The paper proposes a continual learning approach for pre-trained models based on adapters. The approach, called LuCA, adds an adapter-calibrator pair to any intermediate representation computed by each layer. The approach is then applied to a class-incremental setup by using a single adapter-calibrator pair on top of the [CLS] features on the last layer. This application is dubbed TOSCA. TOSCA keeps a separate adapter-calibrator pair per task (like a task-incremental setup). However, TOSCA does not use task labels and relies on confidence prediction among all pairs to select the final answer. The approach shows an effective performance on a range of image classification tasks.

The approach is an application of adapter-based strategies to the final layer of pre-trained models (kept frozen). While I do not particularly enjoy using models pre-trained on ImageNet to study the performance on datasets like CIFAR or ImageNet itself, I acknowledge that this is becoming a common practice in continual learning. The authors also made an effort to test TOSCA on other datasets as well, with good results.
Overall, the main merit of the proposed approach is that it achieves a solid performance against state-of-the-art approaches.

**Audience:**

Yes

**Audience Explanation:**

The topic of continual learning for pre-trained models is an interesting topic for the machine and deep learning community.

**Broader Impact Concerns:**

No concerns.

**Claims And Evidence:**

No

**Claims Explanation:**

The empirical evaluation results show that TOSCA is effective on image classification tasks. I find this claim to be supported by the  discussion in the paper, even though some additional studies could make the point stronger (see comments below).

However, the neuroscientific inspiration for TOSCA, claimed across the whole paper, is simply not present (again, see comments below).

The claim about task-wise plasticity is also debatable: since the pre-trained model is fixed, it would be surprising to observe a plasticity loss. I do not consider the task-wise plasticity a positive consequence of TOSCA.

**Requested Changes:**

1. In the Introduction, the authors claim to "take inspiration from neuroscience" for the design of TOSCA, mentioning the invariant representations developed by the ventral visual stream and the task-specific circuitry of the prefrontal cortex. Subsequent claims of neuroscientific inspiration are spread out across the rest of the paper as well.
I personally don't like such vague, neuroscience-inspired claims when the approach clearly does not really build on any neuroscientific-related knowledge. Properties such as "forming invariant representations"... "while flexibly adapting and modulating them through task-specific circuits" is a broadly applicable statement. I might be wrong on this, but it really looks like the source of inspiration has been devised a posteriori rather than used as a starting point to design the approach. If that is the case, the provided statements are not entirely honest.
Otherwise, are the authors really convinced that a large pre-trained model equipped with an adapter mechanism emulates high-level phenomena exhibited by the ventral visual stream or the prefrontal cortex?
1. In the methodology section, high-level statements like "It decouples feature transformation from discriminative feature enhancement with the dual adapter-calibrator architecture, allowing precise control over parameter updates." are confusing to me. For example, it would be much more informative to start the same paragraph by framing the proposed approach as an adapter-based method.
1. Why is the calibrator called as such? What is the evidence that there is any calibration going on?
1. Is LuCA more efficient in terms of the number of trainable parameters with respect to the adapter approaches previously discussed? It looks like the only advantage is due to the fact that TOSCA only applies LuCA at the [CLS] level. Also, I do not really see the need to introduce two different names for the same approach.
1. There is no analysis on the impact of the soft-attention gating function. Is some sort of attention really happening?
1. There is no analysis on the task-inference performance. Are TOSCA correctly selecting the adapter-calibrator pair related to the inference task? What is the confidence of each module?
1. The theoretical underpinnings do not seem to add much to the overall contribution. The pre-trained model is always frozen, so the feature manifolds cannot change. The second theoretical consideration is related to the change in the feature manifold of the [CLS] token, which is upper-bounded by a small constant, according to the authors. There is no proof or bound on the constant. I do not think it would be that useful even if there were one. Overall, if there are no theoretical considerations to be made, the authors could avoid them altogether. I don't think this would diminish the impact of the work. Trying to create a theoretical derivation out of nowhere might not always be the way to go.
1. The ablation study in Figure 4c should be properly described, as it is very interesting. The authors could consider providing a more detailed experimental setup for each of the settings in the plot.
1. Since the pre-trained model is entirely frozen, why were the authors expecting a task-wise plasticity loss?


Notation:
1. In Equation 2 shouldn't N be M? Also, why is $\phi$ part of the optimization problem? Why is the key $k_{s_i}$ indexed like that if it was previously indexed by $i$ only? What is the difference?
1. Equation 3 is difficult to read. The argmin returns a set of indices but if I understood its meaning, the equation should return a prompt, not a set of indices.
1. What is the difference between equation 7 and eq 5 since $z=\phi(x)$?

Minor changes:
1. Acronyms should be introduced before use: e.g., PEFT in the introduction, N in equation 2.
1. I did not completely get the difference between architectural methods and parameter-isolation methods in the related work section. Aren't they using separate modules? Pruning is often part of architectural methods as well.
1. Equations are detached from the text, and they just interleave the paragraphs. They should be part of the text as if they were words.
8. Caption of Figure 1 should be expanded to be self-contained.

---

> ### Author Response · Authors · 2025-12-30
>
> We sincerely thank the reviewer for their careful review and constructive feedback. We substantially improved our study and address each comment in detail below. For clarity, all revisions in the revised manuscript are highlighted in blue.
>
> **Requested Change 1.** We appreciate the reviewer’s perspective and understand the concern regarding the neuroscience-inspired claims. To clarify, we would like to transparently describe the actual reasoning process and storyline of this paper:
> Our starting point was an observation from the continual learning literature: existing FM-based CIL methods (with the exception of L2P) insert adaptation modules throughout all layers to improve performance. While effective, this strategy is computationally expensive, increases inference overhead, and feels conceptually misaligned with how biological systems operate, appearing as if the entire “brain” must expand for each new task.
> This motivated us to look into relevant neuroscience literature to understand how biological systems reuse representations. The prevalent view is that the ventral visual stream forms robust, largely task-agnostic representations, while more flexible, task-specific adjustments occur only through relatively small circuits closer to the action or decision stage. This observation aligned with our intuition that adaptation in FM-based CIL might be more effective when applied late in the processing pipeline, rather than distributed across all layers.
> Therefore, this neuroscientific framing was not devised a posteriori; rather, it shaped the initial design choice to constrain adaptation to the final stage of the model. As expected, our first implementation used a standard adapter, but as illustrated in Fig. 4(c), this alone was insufficient. This led us to develop a stronger, two-step mechanism (calibration + adaptation), resulting in our new lightweight module, LuCA. Thus, the contribution ultimately consists of two components:
>
> - **(i)** **LuCA**, a more expressive module than a standard adapter, and
> - **(ii)** **TOSCA**, a design strategy motivated by the neuroscience-inspired insight that late-layer modulation can effectively leverage strong invariant features.
>
> We hope this explanation clarifies that our neuroscientific analogy genuinely guided the development process, rather than being added retrospectively.
>
> **Requested Change 2.** Thank you for raising this concern. We have accordingly reframed the LuCA module to more clearly align with an adapter-based formulation stating as, "Our module is based on an adapter-style design, where it learns refined task-specific information."
>
> **Requested Change 3.** We now explain this component more explicitly in the Methodology section. The calibrator $ C(\cdot) $ reweights the adapter’s output features via an attention-like gating mechanism and refines discriminative features according to Eq. (6), where $\odot$ denotes the Hadamard product and $\sigma$ indicates a sigmoid activation function. We refer to this component as a *calibrator* because its gating mechanism performs task-dependent feature calibration: it produces a soft importance mask that rescales each feature dimension based on its relevance to the current task. This operation suppresses over-activated or noisy channels while amplifying stable and discriminative ones, thereby calibrating the representation before it is passed to the classifier.

---

> ### Author Response · Authors · 2025-12-30
>
> **Requested Change 4.**
> LuCA is the general-purpose module that can, in principle, be inserted at any layer or multiple layers of a network. Its design is not restricted to continual learning and can be readily applied to other settings such as transfer learning or general adaptation scenarios. TOSCA, in contrast, is a *specific instantiation* of LuCA tailored to continual post-training: (i) applies LuCA exclusively to the final [CLS] token, (ii) is motivated by neuroscientific insights on late-stage modulation, and (iii) employs per-task instantiation with sparse $ \ell_1 $-regularization. A version of LuCA deployed across all layers would differ substantially *(computationally, parametrically, and conceptually)* from this final-token continual learning formulation. For this reason, a clear naming distinction is necessary to avoid confusion. Notably, applying LuCA to all layers would be unnecessary and would undermine our efficiency goal by over-parameterizing the model; consequently, it would no longer correspond to TOSCA.
>
> **Requested Change 5.** Yes, the calibrator’s gating mechanism performs a form of gated soft attention. Specifically, the calibrator applies a sigmoid activation to the learned residual transformation of the [CLS] features, producing a vector of importance scores in [0, 1]. Each dimension reflects how relevant the corresponding feature is for the current task. These scores are then applied element-wise to the adapter output, amplifying discriminative features while suppressing less informative ones.
>
> **Requested Change 6.** We thank the reviewer for highlighting the importance of evaluating inference-time task identification. In response, we have added a new analysis to the appendix reporting the per-task module-selection success rate under our entropy-based selection policy. TOSCA consistently selects the module with high accuracy across all tasks. These results confirm that the entropy-based criterion is not only theoretically motivated but also highly reliable in practice, with consistently strong module-selection performance.
>
> **Requested Change 7.**  We understand and appreciate the reviewer’s point. Our intention with the theoretical discussion was not to overstate a theoretical contribution, but rather to provide intuition and formal grounding for how our design choices affect representation shifts, particularly since some readers find such framing helpful for interpreting architectural decisions. We agree, however, that the main contributions of the paper are empirical and methodological rather than theoretical. To avoid distracting from the core message, we have moved these theoretical considerations to the appendix in the revised version. This preserves the additional intuition for interested readers while keeping the main paper focused on the practical and conceptual contributions of TOSCA and LuCA.
>
> **Requested Change 8.** We thank the reviewer for guiding us towards an important direction. To provide more insights on different components of the proposed method we have now extended our ablation paragraph and added a new Further Discussion section. Here we explain the effect of different architectural components and to provide more intuition we analyze the shape of the learned representation space by visualizing t-SNE embeddings of the CLS-token features under three configurations: the vanilla [CLS], [CLS] + Adapter, and  [CLS] + TOSCA.
>
> **Requested Change 9.** We appreciate the reviewer’s question. To clarify, here, we would like to quantify the task-wise plasticity or adaptivity of each method. Even with a frozen backbone, different adaptation mechanisms vary in how effectively they can adjust to new tasks through their lightweight modules. Additionally, as the number of tasks increases, module selection becomes increasingly challenging for all approaches. Therefore, evaluating task-wise plasticity helps us understand how well each method can adapt under this growing complexity.
>
> **Notation and Minor Changes.** Thank you for pointing these out. We have corrected the typos and notation errors in the revised manuscript.

---

> > ### Comment · Reviewer_x8Rr · 2026-01-05
> >
> > I thank the authors for their response.
> >
> > I still think the neuroscientific inspiration is loose at best. The neuroscientific processes in question has little to do with the work done by TOSCA. Personally, I would prefer a more cautious and rigorous use of neuroscience (if needed at all) when motivating machine learning methods. That said, this is just my perspective, and the practice is fairly common across many machine learning papers.
> >
> > I appreciate the additional analysis on the calibrator role. Perhaps the experimental setup behind Figure 7 could be better clarified. For example, there is no mention of the dataset used to compute the ViT representations.
> >
> > My main concerns were addressed and I lean towards acceptance.

---

> > > ### Author Response · Authors · 2026-01-14
> > >
> > > We appreciate the reviewer’s positive assessment and are glad that the main concerns have been addressed. We will clarify the experimental setup of Figure 7 in the camera-ready version, noting that it is plotted from the first learning session of CUB B0 Inc10 using ViT-B/16 pretrained on IN-21K.

---

### Review · Reviewer_thQQ · 2025-12-09

**Summary Of Contributions:**

This paper introduces TOSCA, a lightweight PEFT module combining a low-rank adapter and a calibrator, and TOSCA, a continual post-training method that places a single sparse LuCA module only on the final [CLS] token for each task. The idea is to adapt the final semantic representation while keeping the entire pretrained model completely frozen.

**Audience:**

Yes

**Audience Explanation:**

The method is conceptually appealing, easy to implement, and empirically strong.

**Broader Impact Concerns:**

Because the backbone remains frozen, any biases present in the original model are fully preserved and potentially reinforced across tasks. Although each module is small, the accumulation of many per-task modules increases storage, inference cost, and environmental footprint over long task sequences.

**Claims And Evidence:**

Yes

**Claims Explanation:**

They run experiments on different datasets, compare against strong baselines, report multiple seeds, and provide runtime + parameter analyses. The improvements are large and consistent.

**Requested Changes:**

- Can you clarify the claims around L1 regularization promoting “orthogonality” ( 5.6 Ablation Study ) ?
- for the “bounded deviation” (Eq. 11), can you explain it further by explaining what is meant by "bounded deviation"? It will be interesting to see how the residual/calibrator structure enforces this as claimed in the text.
- Can you provide more intuition for the separate roles of the adapter and calibrator, and why the adapter/calibrator ordering performs best? A short conceptual or geometric explanation would make Section 5.6 more insightful
- Can you add a “last-layer fine-tuning” baseline? This is the fairest baseline to compare against TOSCA I think
- Can you explain the figure 4c further in the text ( It's an interesting figure but in the text you didn't analyse it enough )
- How does entropy-based task selection perform when the task boundries are blur ?

---

> ### Author Response · Authors · 2025-12-30
>
> We thank the reviewer for the detailed and constructive feedback. Following the suggestions, we have substantially revised the manuscript and added new experimental results and analyses.
>
> **Requested Change 1.**
> We have elaborated on this mechanism and its implications in the our new Further Discussion section. We also clarify the reviewer’s question here. Our claim regarding ℓ₁ regularization is conceptual rather than a statement of strict mathematical orthogonality. While we do not explicitly enforce orthonormal bases, sparsity effectively promotes disentanglement between task-specific components, a phenomenon also observed in prior sparsity-based continual learning methods. To support this claim, we include an additional study in the revised manuscript where we vary the strength of the ℓ₁ regularization and analyze pairwise cosine similarity between modules (Figure 6). The results show that increasing ℓ₁ strength consistently reduces cosine similarity, indicating reduced overlap in active feature dimensions. Consequently, modules rely on more disjoint subsets of features, leading to lower inter-module correlation and more orthogonal behavior in practice.
>
> **Requested Change 2.**
> In Equation 11, the term *bounded deviation* refers to the controlled nature of the representation change introduced by the adapter–calibrator combination. The adapter contributes an additive residual representation, while the calibrator multiplicatively regulates its magnitude. Formally, if the adapter produces a shift Δh, the calibrator scales this shift by values in [0, 1], yielding $h' = h + c(h) \odot \Delta h $, with $\|h' - h\| \leq \|\Delta h\| $. As a result, the deviation from the original feature representation remains bounded by the learned calibration. Intuitively, the adapter introduces task-specific information, whereas the calibrator suppresses excessive or task-irrelevant deviations, ensuring that the adaptation remains controlled. We have expanded the discussion of the adapter and calibrator components accordingly in the revised manuscript.
>
> **Requested Change 3.**
> Conceptually, the adapter injects task-specific information through an additive residual, while the calibrator selectively suppresses or re-weights this information based on task relevance. Geometrically, the adapter induces a shift of the representation toward a task-specific subspace, after which the calibrator applies class-wise feature contraction within this subspace, thereby improving separability. Applying the adapter before the calibrator provides stronger control, as the representation is first adjusted and then regularized; reversing this order weakens the effect since calibration would operate on unmodified features.
> To further build intuition about the distinct roles of these architectural components, we analyze the learned representation space using t-SNE visualizations of the CLS-token features under three configurations: vanilla [CLS], [CLS] + Adapter, and [CLS] + TOSCA (Figure 7). The progression across these visualizations shows that the adapter increases class separability by introducing task-specific transformations, while the calibrator further sharpens and compresses these clusters, yielding a more discriminative and structured representation space. This geometric progression aligns with the observed performance improvements. A detailed discussion has been added to Section 5.8.
>
> **Requested Change 4.**
> We agree that incorporating last-layer fine-tuning could serve as an informative baseline. However, this setting falls outside the scope of our work, which is positioned as a PEFT approach that are specifically designed to keep the pretrained foundation model entirely frozen and to adapt it using only lightweight learnable embeddings or small auxiliary modules. Moreover, the suggested baseline would require additional mechanisms to mitigate forgetting, as repeatedly fine-tuning the last layer would overwrite previously acquired task information. Without such safeguards, this baseline would not constitute a fair comparison under CL. We believe that the newly added RanPAC closely aligns with the spirit of the reviewer’s suggestion and therefore hope this addition sufficiently addresses the reviewer’s concern.
>
> **Requested Change 5.**
> We have substantially expanded the analysis of Figure 4c in the revised version and add a new discussion section provides both conceptual and geometric intuition.
>
> **Requested Change 6.**
> We agree that entropy-based selection may become less reliable when task boundaries are less pronounced, as routing uncertainty increases and confidence scores become more diffuse. We therefore already acknowledge this limitation in the manuscript. One potential direction to mitigate this issue is an ensemble-based selection strategy, where modules with higher confidence (lower entropy) exert greater influence on the final decision. We view this as a promising avenue for future work.

---

### Review · Reviewer_Qtdv · 2025-12-18

**Summary Of Contributions:**

In this work, the authors propose **LuCA** (*Learn and Calibrate*), a Parameter-Efficient Fine-Tuning module that integrates LoRA-style adaptation weights with an attention gating mechanism. Furthermore, they extend the utility of LuCA to the Continual Learning setting through **TOSCA** (*Token-level Sparse Calibration and Adaptation*). The TOSCA framework trains a LuCA module directly on the $\texttt{[CLS]}$ token, incorporating a sparsity loss to encourage task specificity across individual LuCA modules.

**Audience:**

Yes

**Audience Explanation:**

The well-established setting of Class-Incremental Learning remains a subject of significant interest to a broad research audience. Within this context, TOSCA positions itself as a contribution that enriches the already dense landscape of existing methodologies.

**Broader Impact Concerns:**

The Broader Impact section raises no concerns.

**Claims And Evidence:**

Yes

**Claims Explanation:**

The experimental evaluation is comprehensive in its dataset coverage, and the results demonstrate that the proposed method achieves state-of-the-art performance compared to the reported baselines. While the additional experiments and ablation studies provide valuable insights, their presentation could be refined for greater clarity.

**Requested Changes:**

### **Major Comments**

**1. Experimental Evaluation and Baselines**

While the experimental section presents a robust evaluation across multiple datasets, the comparison would benefit significantly from the following additions:

* **State-of-the-Art Baselines:** In the well-established Class-Incremental Learning (CIL) setting, the absence of **Hide-Prompt** [1] and **RANPAC** [2] represents a gap in the comparative analysis. Hide-Prompt is widely recognized as a state-of-the-art method in the prompt learning category and is cited by the authors; similarly, RANPAC offers a robust analytical approach. Including these methods --- both in the performance tables and the "Memory & computational cost" plot --- is essential for a complete assessment.
* **Inclusion of EuroSAT** [3]: To rigorously assess Out-of-Distribution (OOD) performance, I recommend including EuroSAT, given its distinct nature as a satellite imagery dataset.

**2. Metric Clarification (Table 2)**

Table 2 requires greater clarity regarding the reported metrics. Specifically, explicitly link the results to the specific accuracy metric $\mathcal{A}$ employed.

**3. Ablation Studies (Figure 4c and TOSCA_r)**

The explanation of the ablation study presented in **Figure 4c**, as well as the definition and role of the variant **TOSCA_r**, needs to be expanded. The current description lacks the detail necessary for full interpretation.

[1] Wang, Liyuan, et al. "Hierarchical decomposition of prompt-based continual learning: Rethinking obscured sub-optimality." Advances in Neural Information Processing Systems (2023).

[2] McDonnell, Mark D., et al. "Ranpac: Random projections and pre-trained models for continual learning." Advances in Neural Information Processing Systems (2023).

[3] Helber, Patrick, et al. "Eurosat: A novel dataset and deep learning benchmark for land use and land cover classification." IEEE Journal of Selected Topics in Applied Earth Observations and Remote Sensing (2019).

---

### **Minor Comments**

**1. Terminology: "Post-training"**

The term "post-training" appears misleading in this context. Although TOSCA utilizes a pre-trained backbone --- a well-established practice --- it still involves a training phase for the final LuCA module and the classifier. It would be more accurate to describe this as "adaptation" or "fine-tuning" rather than implying training has ceased.

**2. Notation and Nomenclature**

Several mathematical notations require correction or clarification:

* In $x_i \in \mathbb{R}^D$, the symbol $D$ creates a conflict, as it is previously used to denote Datasets.
* The statement *"Finally, the embedded [CLS] token is considered as $\phi(x)$"* is unclear, particularly following the introduction of $x_e$. While the $\texttt{[CLS]}$ token is a standard concept, the specific mapping from $x_e$ to $\phi(x)$ should be formalized.
* The phrase *"across all encountered classes"* appears contradictory if followed immediately by $\mathcal{Y}_b$ (which usually implies the current task batch). Ensure the scope of the classes (cumulative vs. current) matches the notation.
* In the phrase *"Given an input instance $x$ with label $y \in \mathcal{Y}_b$,"* verify if the intended notation for the set of labels is $\mathcal{Y}_b$ or $Y_b$ for consistency.
* In the section "Learning Prompts," the variable $k_{s_i}$ is introduced without definition. Please clarify the query mechanism used for key matching.

**3. Structural Observations**

* The "Background" section provides explicit equations for prompt learning. Since the proposed method does not utilize prompts, this level of detail is unnecessary and could be condensed to focus on relevant concepts.
* There are multiple instances where equations are cited in the text before they are displayed. This disrupts the reading flow; ensure equations are introduced or displayed prior to, or immediately alongside, their textual reference.

---

> ### Author Response · Authors · 2025-12-30
>
> We thank the reviewer for taking the time to review our paper and for the helpful comments. Below, we address each comment in detail. For ease of review, all revisions in the manuscript are highlighted in blue.
>
> **Major Comment 1.**
> Hide-Prompt and RanPAC have been added to the main paper for completeness. We additionally report performance on the EuroSAT dataset as an out-of-distribution evaluation to assess generalization, where TOSCA consistently demonstrates state-of-the-art performance.
>
> **Major Comment 2.**
> For clarity, we have updated the table caption to explicitly indicate which notation corresponds to each metric. These notations are consistent with the definitions already provided in the evaluation metrics section.
>
> **Major Comment 3.**
> We have strengthened the ablation study and expanded the corresponding discussion. The revised version now includes additional analyses and insights such as t-SNE visualizations, orthogonality estimation, and an evaluation of module selection success, providing a more comprehensive understanding of the behavior of the proposed method.
>
> **Minor Comment 1.**
> We appreciate the reviewer’s perspective and have added a footnote on the first page clarifying that, by *post-training adaptation strategies*, we specifically refer to lightweight adaptation methods built on top of large, frozen, pre-trained backbones.
>
> **Minor Comment 2.**
> We have carefully revised the notation throughout the manuscript to improve clarity and consistency:
>
> - We replaced the symbol to avoid notational conflicts, and now denote the mapping as $\mathbb{R}^{H \times W \times C} \rightarrow \mathbb{R}^{d}$.
> - To clarify the relationship between $x_e$ and the [CLS] token, we rewrote the corresponding paragraph to make this connection explicit.
> - We ensured that $\mathcal{Y}_b$ consistently represents the set of all labels observed so far, now defined as
>   $\mathcal{Y}_b = Y_1 \cup Y_2 \cup \cdots \cup Y_b$.
> - The notation for $\mathcal{Y}_b$ has been updated accordingly throughout the manuscript.
> - We revised the notation for the variable $k_{si}$ and explicitly introduced the prompt selection mechanism for improved readability and understanding.
>
> **Minor Comment 3.**
> We have revised the equations to better integrate it into the surrounding text, ensuring a smoother reading flow without interrupting the narrative.

---

### Author Response · Authors · 2025-12-30

We sincerely thank all reviewers and the Action Editor for their time, careful reading, and constructive feedback. We believe that the manuscript has been substantially improved as a result of these suggestions.

The revised version of the paper is now available. For ease of review, all changes and additions are highlighted in blue throughout the manuscript.

In the revised version, we added two recent state-of-the-art baselines, RanPAC and HiDePrompt, as well as a new out-of-distribution benchmark, EuroSAT, to strengthen the empirical evaluation. We also expanded and improved the ablation study section and introduced a new Further Analysis and Discussion section, where we provide deeper insights into orthogonality and representation space behavior, offering a clearer geometric interpretation of our approach.

Additionally, we included an evaluation of the module selection performance to demonstrate its practical effectiveness. Finally, we refined the notation, corrected minor issues, and improved the overall flow and readability of the paper.

---

### Decision · Action_Editor_rCr7 · 2026-01-22

**Recommendation:** Accept as is

**Audience:**

Yes

**Audience Explanation:**

Researchers working in the field of continual learning will find this paper interesting and useful.

**Claims And Evidence:**

Yes

**Claims Explanation:**

This paper proposes a simple parameter-efficient fine-tuning method for class incremental learning called TOSCA. All the reviewers agree that the work is simple, efficient, and very useful for the community.